# What to consider before prescribing inhaled medications: a pragmatic approach for evaluating the current inhaler landscape

*Ther Adv Respir Dis*

2019, Vol. 13: 1–28

DOI: 10.1177/
1753466619884532

© The Author(s), 2019.

**Federico Lavorini** (ID)**, Christer Janson, Fulvio Braido, Georgios Stratelis and Anders Løkke**

*Abstract*: Inhaled therapies are the cornerstone of treatment in asthma and chronic obstructive pulmonary disease, and there are a multitude of devices available. There is, however, a distinct lack of evidence-based guidance for healthcare providers on how to choose an appropriate inhaler. This review aims to summarise recent updates on topics related to inhaler choice, and to offer practical considerations for healthcare providers regarding currently marketed devices. The importance of choosing the right inhaler for the right patient is discussed, and the relative merits of dry powder inhalers, pressurised metered dose inhalers, breath-actuated pressurised metered dose inhalers, spacers and soft mist inhalers are considered. Compiling the latest studies in the devices therapy area, this review focuses on the most common types of handling errors, as well as the comparative rates of incorrect inhalation technique between devices. The impact of device-specific handling errors on inhaler performance is also discussed, and the characteristics that can impair optimal drug delivery, such as inhalation flow rate, inhalation volume and particle size, are compared between devices. The impact of patient perceptions, behaviours and problems with inhalation technique is analysed, and the need for appropriate patient education is also highlighted. The continued development of technology in inhaler design and the need to standardise study assessment, endpoints and patient populations are identified as future research needs.

*The reviews of this paper are available via the supplemental material section.*

*Keywords:* asthma, COPD, critical errors, dry powder inhaler, inhalation technique, metered dose inhaler

Correspondence to:
**Federico Lavorini**
Careggi University
Hospital, Department of
Experimental and Clinical
Medicine, Largo Brambilla
3, 50134, Florence, Italy
**federico.lavorini@unifi.it**

**Christer Janson**
Department of Medical
Sciences: Respiratory,
Allergy and Sleep
Research, Uppsala
University, Akademiska
sjukhuset, Uppsala,
Sweden

**Fulvio Braido**
Allergy and Respiratory
Disease Clinic, DIMI-
University of Genova,
IRCCS AOU San Martino-
IST, Genova, Italy

**Georgios Stratelis**
Department of Medical
Sciences: Respiratory,
Allergy and Sleep
Research, Uppsala
University, Akademiska
sjukhuset, Uppsala,
Sweden

AstraZeneca Nordic-
Baltic, Astraallén,
Södertälje, Sweden

**Anders Løkke**
Department of Respiratory
Diseases and Allergy,
Aarhus University Hospital,
Aarhus C, Denmark

## Introduction

Asthma and chronic obstructive pulmonary disease (COPD) are serious public health concerns associated with high levels of chronic morbidity and mortality throughout the world.[1,2] Despite the availability of effective medications and management strategies,[3,4] they continue to represent a large global burden.[5] Clinical strategies for the management of asthma and COPD recommend that inhaled therapy forms the cornerstone of the treatment of both diseases.[3,4] With the need for effective inhaled drug delivery comes a variety of devices that allow for rapid and targeted delivery of the therapeutic agent(s) to the lungs, providing a high concentration at the site of action and low systemic exposure.[6,7] However, while the efficacy

and safety of the various inhaled agents and drug combinations is an obvious consideration for healthcare providers when choosing appropriate therapy for a patient, the choice of device is also a vital factor; a factor for which there exist no regulatory preferences,[8] and current clinical strategies provide little guidance.[9]

## Initial considerations for healthcare providers when selecting an inhaler device

The wide selection of available devices allows treatment to be tailored to the individual patient, but also complicates the decision on which device to choose.[10] Each type of inhaler requires a particular inhalation technique, which, if not mastered, can

negatively affect drug delivery to the lungs. Alongside drug properties, key factors to consider include the inspiratory profile that a patient can achieve, as well as age, cognitive capacity and functional ability of the patient.[8,11,12] The Aerosol Drug Management Improvement Team (ADMIT), an expert group of European respiratory physicians, states that the primary factors governing device selection should be efficacy and safety, but the limited clinical data from direct head-to-head trials was highlighted as a concern.[13] They proposed the factors listed in Table 1 as key considerations for device selection. Patient acceptance of devices is also potentially important, as patient surveys have shown disparity in patients' opinions of different devices;[14] less-favoured devices may correlate with poor adherence,[15] and characteristics such as usability and ease of training could affect this, although this is often difficult to determine.[16]

The importance of physician familiarity with a device has also been highlighted.[10] A Delphi consensus statement detailed some of the inhaler device selection factors judged to be most critical by experts in asthma and COPD:[17] consensus was achieved for the choice of inhalation device being as important as that of the active substance. For criteria influencing choice of inhaler device, patients' ability to use the device correctly scored highest with physicians. Another Delphi consensus survey of COPD-expert pulmonologists also resulted in near total agreement (98%) that the ability of the patient to handle the inhaler was a relevant factor, with 90% agreeing that a patient's previous experience with an inhaler should be considered.[18] Three-quarters of respondents thought that the physician's own experience with the inhaler was relevant, and patient and physician preference were considered relevant by 71% and 54%, respectively.[18] Ease of teaching and ease of use also scored highly in the consensus statement, and there was a good level of agreement against physicians just prescribing the least costly device.[17] However, it should be noted that such economic recommendations should be considered with regard to the cost pressures facing patients in certain regions, and their effects on adherence; recent research by Laba and colleagues, for example, has shown that discussions between doctors and patients are needed to try and minimise cost-related underuse of respiratory medicines.[19]

The Real-life Experience and Accuracy of inhaLer use (REAL) survey evaluated real-world data on commercially available inhalers delivering COPD maintenance therapy to identify attributes that influence patient adherence and optimal inhaler use.[15] Self-reported information from 764 patients showed that age $\leqslant 65$ years significantly reduced patient adherence (69% *versus* 78% for patients aged $>65$ years; $p = 0.02$). The effect of training on device use was clear, in that trained respondents were significantly more confident in their inhaler use ($p = 0.001$), with demonstrative training by an expert shown to be particularly effective. Patients considered fully adherent (i.e. those taking their medication every day in the previous 30 days) varied from 58% to 90% between the different devices studied, thus underlining the importance of appropriate inhaler selection.

The objective of this review is to summarise the latest available data related to choice of inhaled asthma and COPD medications from a healthcare provider perspective, and to provide practical considerations regarding the available devices.

## Methodology

### Search strategy and selection criteria

The authors conducted an electronic search of the literature using Medline® and Medline® In-Process (PubMed), and Embase in Ovid® (Ovid Technologies, Inc., New York, NY, USA). Criteria for article inclusion were publications from 2010 to April 2018, English language-only publications, publications limited to original research and systematic reviews. Relevant records were identified using the search terms 'Asthma' or '(COPD or 'chronic obstructive pulmonary disease')' along with at least one device type or brand name and at least one keyword or phrase outlined in supplementary Table S1.

Any duplicate articles were removed during the electronic search process, and other selected papers deemed to be of key interest but which were outside of the agreed date range were added. Search results were reviewed by title and abstract to identify any possibly or definitely relevant papers. In this process, results were also separated into the following categories: device characteristics, device handling and storage errors, technical factors to consider, patient education, and technology and future innovations (level 1 filtering). Results were not filtered by disease type or severity. Key papers

**Table 1.** Considerations for appropriate device selection.[10,13]

| **Factors influencing selection of delivery device** |
| --- |
| Efficacy and safety |
| Clinical setting |
| Economic constraints |
| Patients' ability to use the prescribed inhaler |
| Age of patient |
| Patients' acceptability of the device |
| Patient preference |
| Pulmonary deposition |
| **Useful questions to facilitate the choice of drug/device combinations for an individual patient** |
| Can the same type of device be used for all inhaled drugs prescribed for the patient? |
| Is the patient able to use the device correctly? |
| Which devices are available that deliver the desired drug? |
| What are the storage restrictions up to and after first use (storage temperature and in-use shelf-life)? |
| Which devices are the most convenient and portable for the patient? |
| With which devices is the physician familiar? |

were identified, and full-text versions were obtained so that the article could be reviewed in depth (level 2 filtering). Filtering was undertaken by professional medical writers with several years of experience in the respiratory therapeutic area.

## Results

Of the 1145 records identified, 352 were initially selected as potentially being of interest. Further manual filtering (level 3 filtering) reduced this figure substantially. Upon full review, some references were deemed not to be relevant and were excluded. Relevant studies from the reference lists of the retrieved publications were added, along with certain papers that fell outside of the search parameters but were deemed to be highly relevant.

## Characteristics of available devices

### The main differences between available device types

Available inhalation devices include dry powder inhalers (DPIs); pressurised metered dose inhalers

(pMDIs), used with or without a spacer; soft mist inhalers (SMIs) and nebulisers [please note: nebulisers will not be covered as part of this review, as they are not recommended for long-term treatment by asthma (GINA)[3] or COPD (GOLD)[4]]. pMDIs and SMIs are classified as propellant-driven aerosol generation inhalers. Most new launches of inhaled medications tend towards tying the drug to a particular inhaler through the use of bespoke devices.[8] Each device type is associated with advantages and disadvantages, and these are outlined below and summarised in Table 2. Useful websites providing advice on available devices and correct inhaler technique include www.asthma.org.uk/advice/inhaler-videos, www. inhalers4u.org/index.php/instructions and www. atemwegsliga.de/correct-inhalation.html.

### pMDIs

First introduced in the 1950s, pMDIs now use a hydrofluoroalkane (HFA) propellant, which replaced the banned chlorofluorocarbon (CFC) propellant in early devices.[6,20] The velocity of the HFA spray is slower than that of the CFC,

**Table 2.** Common errors and problems associated with the different device types.[6,8,11,13,20–22]

| Device type | Error/problem |
|---|---|
| All device types | Not removing cap/cover fully before inhalation<br>Specific preparation steps are required for effective inhalation and drug delivery<br>Some devices require a specific priming procedure<br>Potential reduction in performance towards the end of the device lifespan<br>Failure to exhale fully before inhalation, or exhalation directly into mouthpiece<br>Failure to inhale fully after starting the inhalation<br>A seal is required to be formed with the lips around the mouthpiece<br>The air inlet(s) may be blocked during inhalation<br>Need to be held in the correct position during dose preparation or inhalation<br>Potential for oropharyngeal deposition if inhalation is too fast<br>No breath-hold following inhalation, or breath-hold may be insufficient<br>Failure to check the number of doses<br>Inhalation despite dose counter at zero<br>Patient may not check the counter has decremented after inhalation<br>No dose, or more than one dose, actuated during inhalation |
| pMDI | Problems with co-ordination of actuation and inhalation<br>Shaking before each actuation required for suspension pMDI (most pMDIs)<br>Not inhaling soon enough after activating the device<br>Failing to place device in mouth (open mouth technique)<br>Requires a slow and deep inhalation, which patients can struggle with<br>Associated with high oropharyngeal deposition<br>May require wet cleaning |
| BA-pMDI | Still requires slow and steady inhalation, as with regular pMDI<br>Still associated with high oropharyngeal deposition, depending on the flow rate (too slow or too fast)<br>Limited availability for many drugs |
| pMDI + spacer | Less portable than a pMDI alone<br>Slow and steady inhalation still required<br>Potential accumulation of electrostatic charge that can affect drug delivery<br>Can reduce the pMDI dose output to a variable extent<br>Require periodic cleaning for optimal functioning<br>Represent an extra cost to using a pMDI alone |
| DPI | Minimum inspiratory flow is required to disaggregate drug particles, although this may substantially differ between devices<br>Some are susceptible to high levels of ambient humidity<br>Susceptible to shaking before or after dose preparation<br>A few DPIs require shaking before dose preparation<br>Some require a number of steps to be performed for preparation and usage |
| SMI | Less availability than DPIs/pMDIs<br>Problems with co-ordination of actuation and inhalation<br>Requires a slow and deep inhalation, which patients can struggle with<br>Can be complex to prepare for use<br>Aseptic manufacturing required, or a preservative added |

Ba-pMDI, breath-actuated pressurised metered dose inhaler; DPI, dry powder inhaler; pMDI, pressurised metered dose inhaler; SMI, soft mist inhaler.

reducing the need for a spacer (discussed below).[6] Some HFA-pMDIs contain solutions and not suspensions, and therefore do not require shaking prior to actuation.[20] Efficient aerosol delivery to the lungs requires slow, deep and steady inhalation (i.e. at a rate of ≤60 l/min for approximately 5 s) starting just prior to device activation, with a subsequent short breath-hold of up to 10 s.[6,11,20,21]

Correct use of the pMDI involves holding the inhaler in the correct position and performing a series of coordinated steps, and the complexity of

this process can prove a challenge to some patients. As mentioned, suspension pMDIs also need to be shaken before use; a step commonly overlooked by both patients and healthcare professionals.[20] As well as problems with timing, many patients struggle to generate a deep enough inhalation,[11] inhale too fast,[6] or fail to hold their breath for long enough.[20]

To overcome the problems associated with poor actuation-inhalation technique, spacers, valved holding chambers (see below) and breath-actuated pMDIs (BA-pMDIs) are available.[6,13] The latter devices are useful for patients who struggle to time their inspiration properly, as they are triggered by airflow upon inspiration,[20] although they still require an inspiratory flow rate of approximately 30 l/min and do not overcome the other disadvantages associated with pMDIs.[20,21]

### Spacers and valved holding chambers

Spacers can be added to a pMDI to overcome problems with coordination, and in doing so help to increase aerosol delivery to the peripheral airways.[6,20,22] Spacers that feature a one-way inspiratory valve are termed valved holding chambers.[6,23] Spacers and valved holding chambers can increase pulmonary deposition compared with pMDIs alone by reducing the velocity of the aerosol and filtering out larger, nonrespirable particles. Slower inspiratory flow rates reduce inertial impaction in the oropharynx and increase deposition in the peripheral airways through gravitational sedimentation and diffusion.[6,20,23] There are a number of recommendations for the optimal inhalation technique with a spacer, which vary depending on the particular devices being used.[23]

### DPIs

Much like the pMDI, DPIs are small, portable and widely available as either single-dose or multiple-dose devices.[6,20,21] DPIs are breath-actuated and require the user to inhale rapidly and forcibly, with a subsequent breath-hold similar to that of pMDIs.[6,20] All DPIs require a pre-inhalation dose-loading step to be completed successfully in order for them to function correctly.[20,22] Because the patient's own inspiratory force drives the drug delivery, unlike the pMDI there is no need to coordinate actuation with inhalation, making DPIs relatively simple to use for the majority of patients.[6,13,21]

A limitation of DPIs is their reliance on patients generating the necessary inspiratory force to ensure effective drug delivery. Most DPIs are formulated with their drug particles attached to excipient carrier molecules such as lactose, although some are in the form of agglomerated pellets.[6] Consequently, DPIs are designed with an internal resistance that must be overcome by a forceful inhalation in order to generate a turbulent flow, de-aggregate the drug particles within, and produce fine particles for inspiration.[6,11,13,20,21]

### SMIs

There is currently only one commercially available SMI, the Respimat® Soft Mist™ Inhaler (Boehringer Ingelheim, Ingelheim am Rhein, Germany).[6] Soft mist inhalers, which by definition are sprays similar to pMDIs, atomise the drug-containing droplets and deliver them as a slower-velocity fine mist compared with ordinary spray devices.[6,22] The longer spray duration is intended to reduce the coordination required between actuation and inhalation compared with pMDIs, but patients are still required to coordinate the actuation and inhalation steps. Actuating the dose too late can result in aerosol being delivered after inhalation has stopped, and a rapid, forceful inhalation may result in the aerosol persisting for longer than the inhalation time.[8,22]

## Technical factors to consider when selecting an inhaler device

### The effect of particle size: theoretical considerations

The aerosol particle size is an important consideration, as particles around 0.5 µm and below in diameter may be exhaled or quickly absorbed into the systemic circulation following deposition in the alveoli, and particles >5 µm can be deposited in the oropharynx and swallowed before ever reaching the lung.[24] Key factors that affect lung deposition include the fine particle dose (FPD), fine particle fraction (FPF) and the mass median aerodynamic diameter (MMAD). The FPD is the absolute mass of particles <5 µm in the total delivered dose (DD), and plays a significant role in the relative distribution of an inhaled drug within the airways.[25] The FPF is the FPD divided by the total DD, and reflects the delivery efficiency of the inhaler. The MMAD is a measure of

the mid-range diameter of particles in a formulation. Aerosols with high FPF or FPD, or low MMAD, are highly likely to penetrate beyond the upper airways and deposit in the lungs, and most current devices generate a considerable proportion of particles in this range.[6] Altering the pattern of particle size distribution, even within this small range, can influence the deposition characteristics of an aerosol.[26]

### The effect of particle size: real-world studies

Several studies and reviews have discussed the potential practical benefit of 'extra-fine' particles in the treatment of asthma and COPD, although clinical data are generally mixed. A systematic review and meta-analysis compared small- and standard-sized-particle inhaled corticosteroid (ICS) formulations (defined as <2 μm and 2–5 μm, respectively) for effects on lung function, symptoms, use of rescue medication and safety in patients with asthma.[26] A total of 23 trials were included but neither the literature review nor the meta-analysis found any significant effect of particle size on efficacy and safety outcomes. The authors concluded that the results did not support the suggestion that smaller size particle ICS are intrinsically more 'effective' than larger, standard size particle ICS on the endpoints of lung function, asthma symptoms and rescue medication use. This study is referenced in the 2019 GINA guideline, which states that 'there is currently insufficient good quality evidence to support use of extra-fine particle ICS aerosols over others'.[3] A review by ADMIT concluded that small-particle (<2 μm) aerosols improve drug deposition and regional airway distribution within the lungs. Small-particle aerosols were more effective in this regard than large-particle (>2 μm) aerosols in real-world studies, but only had comparative efficacy in clinical trials.[25] Experimental and extrapolated *in vitro* deposition data for various particle sizes found that the deposition fraction for particles in the submicron range (diameter <1 μm) is gradually decreased as the particles get smaller compared with those in the micron range (1–5 μm).[24] The range from 1.25 to 3.5 μm achieved the highest amount of lung deposition compared with the exhalation and oropharyngeal deposition percentages at low to moderate flow rate.[24,27] In a matched cohort study of data from the PHARMO Database Network, extra-fine-particle ICS formulations were associated with better odds of asthma control than fine-particle ICS formulations, and this was achieved at a substantially lower prescribed dose.[28]

This finding is also supported by a meta-analysis of observational studies which demonstrated that extra-fine ICS formulations have significantly higher odds of achieving asthma control with lower exacerbation rates than fine-particle ICS formulations.[29] However, the included studies had considerable methodological heterogeneity and variable adjustment for confounding factors. A 48-week randomised, parallel-group study of extra-fine-particle beclometasone/formoterol (metered dose: 200/12 μg twice daily *via* pMDI) compared with budesonide/formoterol (metered dose: 400/12 μg twice daily *via* Turbuhaler® DPI) found that increases in predose morning $FEV_1$ were comparable ($p = 0.93$).[30] The mean rate of exacerbations was also similar between treatments, but the number of patients with COPD exacerbations leading to hospitalisation was significantly lower with budesonide/formoterol (2.9% *versus* 5.6%; $p < 0.001$).

### Inhalation flow rate

A key characteristic of DPIs is the dependency of the FPD on the inhalation flow rate.[31,32] The force required to create a turbulent energy and generate an aerosol is the product of patient inhalation flow and the internal resistance of the device:[6,33]

$$\sqrt{P} = Q \text{ x } R$$

Where $p$ is the change in pressure (turbulent energy), Q is the inhalation flow and R is the inhaler resistance.[34] Subsequent lung deposition is a trade-off between generating sufficient power for particle de-aggregation whilst avoiding the increased oropharyngeal deposition that can occur at higher aerosol velocities.[32]

The ability of certain patient populations to generate the required peak inspiratory flow (PIF) can impact an inhaler's efficacy, as a low PIF can reduce the FPD delivered by over 50%.[35] Children and the elderly with asthma, and patients with COPD or neuromuscular disease are particularly at risk of this.[35,36] However, it should be noted that reduced PIF may already be a confounding variable in some clinical studies involving these particular patient groups, making the magnitude of effect harder to elucidate.

The PIF values for Turbuhaler® (AstraZeneca) and Diskus® (GlaxoSmithKline; branded as Accuhaler® in the United Kingdom and Spain)

were compared in asthma and COPD patients aged >60 years with those aged ≤60 years.[37] The PIF generated by the older group was significantly lower than that in the younger population when using Turbuhaler® (*p* = 0.01) but not with Diskus® (*p* = 0.86), which was attributed to the higher intrinsic resistance of the Turbuhaler® device. The study did not measure the effect of this on clinical outcomes, however, and previous studies have shown that children and patients with acute asthma and severe COPD were able to generate satisfactory inspiratory flow rate (>30 l/min) with Turbuhaler®.[38–40] Interestingly, another study of the Diskus® DPI found that measurement of spirometric PIF can be used as a surrogate to estimate the PIF a patient is likely to generate while using the device.[35] Spirometric PIF cut-offs of <196 l/min and 115 l/min corresponded to a Diskus® PIF of 60 l/min (optimal delivery) and 30 l/min (minimum required), respectively. Therefore, it was concluded that spirometric PIF could be used to inform decisions about patient suitability for Diskus® DPI. It was suggested that this approach may also have value for the use of other flow rate-dependent DPIs; however, this remains to be studied, which may prove difficult due to the varying intrinsic resistance of the devices.

A retrospective study evaluated the impact of PIF on readmission after hospitalisation for acute COPD exacerbations and a subsequent prescription for a DPI.[41] Suboptimal PIF, defined in this study as ≤60 l/min, was present in 52% of patients, and in 60% of patients aged >65 years. It was also predictive of 90-day readmission for COPD. The all-cause and COPD 30- and 90-day readmission rates were significantly lower for those discharged with a nebuliser compared with a DPI (*p* ≤ 0.011).

The currently available DPI devices have varying internal resistance to air flow, which can be classified by the inhalation flow required to produce a 4 kPa pressure drop.[6] The inhalation characteristics of several devices, as well as some other important technical parameters to consider, are presented in Table 3. An *in vitro* study compared the FPD dependency of the inhalation flow rates between Turbuhaler®, Spiromax® (Teva Pharmaceutical Industries, Petah Tikva, Israel) and Easyhaler® (Orion Corporation, Espoo, Finland) for budesonide/formoterol delivery.[31] The FPD ratios of low *versus* medium flow and high *versus* medium flow were similar for all

inhalers and strengths, and for both components. FPD for the budesonide component was consistent between inhalers and within strengths, but FPD for formoterol was consistently higher with Turbuhaler® compared with the other devices. The authors ultimately concluded that the devices tested were equally flow-dependent with regards to the FPD of budesonide and formoterol. However, as the dependency of FPD on the inhalation flow rate is a fundamental characteristic of DPIs, it was noted that the magnitude of decrease in FPD for some of the inhalers tested may have clinical implications in patients with low inhalation capacity or those not following inhaler instructions.[31]

Although pMDIs require slow inhalation, there also needs to be a minimum flow rate. Flow rates for pMDIs should be ≥15 l/min for optimal performance (roughly corresponding to a total inhalation time of 4–5 s), and the lower limit for an acceptable flow rate should be 10 l/min.[80] There are minimal data for minimum inspiratory flow rates when a spacer is used, but a flow rate of >15–20 and <30 l/min is generally recommended.[80]

*Inhalation volume*
The inhalation volume required for complete dose emission is an important factor to consider, as a low volume could have a negative effect on the DD.[31] The DD from an inhaler depends on an effective discharge (dose emission) from the device during inhalation. In order for peripheral deposition to occur, the discharge must be completed in a fraction of a second and within the first litre of inhaled air.[81] However, the European Pharmacopoeia have recommended that the *in vitro* aerodynamic dose emission characteristics emitted from a DPI are measured using a constant inhaled volume of 4 l and a resultant pressure drop of 4 kPa.[82] Therefore, at a test volume of 4 l, an inhaler with a DD that was volume-dependent would not be detected, nor would this property be disclosed in the regulatory file, as these data are not required by the regulatory authorities to be included in the drug product characteristics section of the licence application. This should be borne in mind when selecting an inhaler based on *in vitro* data of this nature.

In an *in vitro* study, dose delivery was assessed for two strengths of budesonide/formoterol administered by Turbuhaler®, Easyhaler® and

**Table 3.** Characteristics of the most common inhaler devices in Europe.

| Characteristics | | Dry powder inhalers (product example) | | | | | | | | | |
|---|---|---|---|---|---|---|---|---|---|---|---|
| | | Turbuhaler® (Symbicort®)[42] | Diskus® (Seretide®)[43] | Breezhaler® (Ultibro®)[44] | Easyhaler® (Bufomix®)[45] | Ellipta® (Relvar®)[46] | Spiromax® (Duoresp®)[47] | HandiHaler® (Spiriva®)[48] | Genuair® (Duaklir®)[49] | Novolizer® (Budelin®, Novopulmon®)[50] | NEXThaler® (Foster®)[51] |
| Drug component(s) | | Budesonide/formoterol | Fluticasone propionate/salmeterol | Glycopyrronium (mono)/indacaterol (mono) | Budesonide/formoterol | Fluticasone furoate/vilanterol | Budesonide/formoterol | Tiotropium | Aclidinium (Eklira®) | Budesonide | BDP/formoterol |
| Multi-dose/single unit dose | | Multi-dose | Multi-dose | Single unit dose | Multi-dose | Multi-dose | Multi-dose | Single unit dose | Multi-dose | Multi-dose | Multi-dose |
| Premetered/reservoir | | Reservoir | Blister strip | Capsules | Reservoir | Blister strip × 2 | Reservoir | Capsules | Reservoir | Reservoir | Reservoir |
| Formulation | | Soft agglomerates | Lactose carrier | Lactose carrier | Lactose carrier | Lactose carrier | Lactose carrier | Lactose carrier | Lactose carrier | Lactose carrier | Lactose carrier |
| Lactose mg/dose (highest–lowest strength) | | None | 12 | 24 | 4–8 | 25 | 5–10 | 6 | 11 | 11 | 10 |
| Excipients other than lactose | | None | None | Magnesium stearate | None | Magnesium stearate | None | None | None | None | Magnesium stearate |
| Inspiratory flow rate at 4 kPa (l/min) | | 59[24] | 75[24] | 103[52] | 57[33] | 70[53] | 63[54] | 39[52] | 64[55] | 77[56] | 55[57] |
| Inspiratory resistance, R (Pa$^{0.5}$ l$^{-1}$ s), calculated from R = √(4kPa)/flow rate at 4 kPa | | 64 | 51 | 37 | 66 | 54 | 60 | 97 | 59 | 49 | 69 |
| FPF at 4 kPa (% of delivered dose) | | 62 (ICS) 63 (LABA)[57] | 23 (ICS) 40 (LABA)[58] | 45 (ICS) 45 (LABA)[59] | 43 (ICS) 41 (LABA)[31] | 25 (ICS) 37 (LABA)[60] | 47 (ICS) 45 (LABA)[31] | 27 (LAMA)[61] | 46 (LAMA)[55] | 41 (ICS)[31] | 57 (ICS) 55 (LABA)[57] |
| MMAD (μm) | | 2.1 (ICS) 2.1 (LABA)[57] | 3.9 (ICS) 4.1 (LABA)[58] | 2.7 (LAMA)[62] 2.6 (LABA)[59] | 2.8 (ICS) 2.7 (LABA)[31] | 3.2 (ICS) 1.8 (LABA)[60] | 2.2 (ICS) 2.2 (LABA)[31] | 3.2 (LAMA)[61] | 2.2 (LAMA, mono)[55] | 2.0 (ICS)[31] | 1.1 (ICS) 1.7 (LABA)[57] |
| Moisture protection (before patient use) | | Tight cap + desiccant | Pouched | Capsules in blisters | Pouched | Pouched | Pouched | Capsules in blisters | Pouched | Pouched | Pouched |
| Storage conditions | Temp. | No special storage restriction | Do not store above 30°C | Do not store above 30°C | Do not store above 25°C | Do not store above 25°C | Do not store above 25°C | Do not store above 25°C | No special storage restriction | No special storage restriction | Do not store above 25°C |
| | Humidity in-use | No special humidity restriction | No special humidity restriction | Use capsule immediately | Protect from moisture | Moisture sensitive | No special humidity restriction | No special humidity restriction | No special humidity restriction | Moisture sensitive | No special humidity restriction |
| Shelf-life | Unopened package | 3 years | 2 years | 2 years | 2 years | 2 years | 3 years | 2 years | 3 years | 3 years | 3 years |

*(Continued)*

**Table 3.** (Continued)

| Characteristics | | Dry powder inhalers (product example) | | | | | | | | | |
| --- | --- | --- | --- | --- | --- | --- | --- | --- | --- | --- | --- |
| | | Turbuhaler® (Symbicort®)[42] | Diskus® (Seretide®)[43] | Breezhaler® (Ultibro®)[44] | Easyhaler® (Bufomix®)[45] | Ellipta® (Relvar®)[46] | Spiromax® (Duoresp®)[47] | HandiHaler® (Spiriva®)[48] | Genuair® (Duaklir®)[49] | Novolizer® (Budelin®, Novopulmon®)[50] | NEXThaler® (Foster®)[51] |
| | Max time in-use | As per expiry date | As per expiry date in EUR 4 weeks (US) | As per expiry date | 4 months | 6 weeks | 6 months | 9 days (blister) | 2 months | 6 months | 6 months |
| | No of inhalations/dose | 1 | 1 | 1–2 | 1 | 1 | 1 | 2 | 1 | 1 | 1 |
| | Inhalation volume dependent delivery (yes for capsule inhalers relates to patient instruction leaflet) | No | No | Yes | No | No | Yes | Yes | No | Yes (emitted mass)[63] | No |
| | Shake/no shake before use according to SPC | N/A | N/A | N/A | Shake | N/A | Do not shake | N/A | N/A | N/A | N/A |
| | Double dosing protection | Yes | Some risk | Yes | No | Yes | Yes | Yes | Yes | Yes | Yes |

| Characteristics | Pressurised metered dose inhalers | | | | | | | | |
| --- | --- | --- | --- | --- | --- | --- | --- | --- | --- |
| | Innovair® (Fostair®)[64] | Symbicort®[65] | Seretide® Evohaler®[66] | Spiolto® Respimat®[67] | Flutiform®[68] | Flutiform®, K-haler®[69] | Alvesco®[70] | Qvar®[71] | Qvar® Easi-Breathe®[72] |
| Drug component(s) | BDP/formoterol | Budesonide/formoterol | Fluticasone propionate/salmeterol | Tiotropium (mono product) | Fluticasone propionate/formoterol | Fluticasone propionate/formoterol | Ciclesonide | BDP | BDP |
| Breath-actuated inhaler | No | No | No | No | No | Yes | No | No | Yes |
| Solute/solvent | Solution | Suspension | Suspension | Solution | Suspension | Suspension | Solution | Solution | Solution |
| Preservative | N/A | N/A | N/A | Benzalkonium chloride | N/A | N/A | N/A | N/A | N/A |
| Propellant or water | Norflurane (HFA-134a) | Apaflurane (HFA-227) | Norflurane (HFA-134a) | Water | Apaflurane (HFA-227) | Apaflurane (HFA-227) | Norflurane (HFA-134a) | Norflurane (HFA-134a) | Norflurane (HFA-134a) |
| Excipients | Ethanol Hydrochloric acid | Povidone Macrogol 1000 | None | Benzalkonium chloride Disodium edetate | Sodium cromoglicate Ethanol | Sodium cromoglicate Ethanol | Ethanol | Ethanol | Ethanol |

*(Continued)*

**Table 3.** (Continued)

| Characteristics | | Pressurised metered dose inhalers | | | | | | | | |
|---|---|---|---|---|---|---|---|---|---|---|
| | | Innovair® (Fostair®)[64] | Symbicort®[65] | Seretide® Evohaler®[66] | Spiolto® Respimat®[67] | Flutiform®[68] | Flutiform®, K-haler®[69] | Alvesco®[70] | Qvar®[71] | Qvar® Easi-Breathe®[72] |
| Shake before use according to SPC | | No | Yes | Yes | No | Yes | Yes | No | No | Yes |
| No of actuations/dose | | 1 | 2 | 2 | 2 | 2 | 2 | 1 | 1 | 1 |
| FPF at 30 l/min (% of delivered dose) | | 39 (ICS) 37 (LABA)[73] | 62 (ICS) 70 (LABA)[74] | 39 (ICS) 38 (LABA)[75] | 63 (LAMA)[76] | 41 (ICS) 39 (LABA)[77] | Not available | 75[78] | 58[79] | Not available |
| MMAD (µm) | | 1.4 (ICS) 1.5 (LABA)[73] | 3.5 (ICS) 3.3 (LABA)[74] | 2.9 (ICS) 3.4 (LABA)[75] | 4.2 (LAMA)[76] | 3.5 (ICS) 3.5 (LABA)[77] | Not available | 1[78] | 1.1[79] | Not available |
| Moisture protection (before patient use) | | No | Pouched | No | N/A | Pouched | Pouched | No | No | No |
| Shelf-life | Unopened package | 15 months | 2 years | 2 years | 3 years | 2 years | 2 years | 3 years | 3 years | 3 years |
| | Max time in-use | 5 months | 3 months | As per expiry date | 3 months | 3 months | 3 months | As per expiry date | As per expiry date | As per expiry date |
| Storage conditions | | Prior to dispensing refrigerator [2°C–8°C] 15 months - After dispensing, 5 months at a temp. up to 25°C | No special storage restriction | Do not store above 25°C | Do not freeze | Do not store above 25°C | Do not store above 25°C | No special storage restriction | Do not store above 25°C, protect from frost and sunlight | Do not store above 25°C |

Parameters not specifically referenced have been taken from the respective product Summary of Product Characteristics.
BDP, beclometasone dipropionate; FPF, fine particle fraction; HFA, hydrofluoroalkane; ICS, inhaled corticosteroid; LABA, long-acting β₂-agonist; LAMA, long-acting muscarinic antagonist; MMAD, mass median aerodynamic diameter.

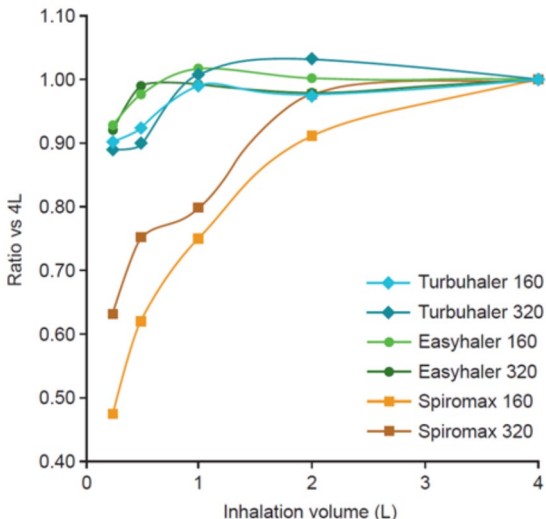

**Figure 1.** Turbuhaler®, Easyhaler® and Spiromax® budesonide delivered dose ratios with various inhalation volumes compared with 4 l for the budesonide/formoterol 160/4.5 and 320/9 μg delivered dose strengths.[31]

Spiromax®.[31] Compared with the results obtained at a simulated inhalation volume of 4 l, DD was unaffected by inhalation volume down to 1 l with Turbuhaler®, with significant differences only for the higher strength at 0.5 l, and both strengths at 0.25 l (all *p* < 0.0125). Easyhaler® DD was independent of inhalation volume down to 0.5 l, with a significant 10% decrease at 0.25 l for the formoterol component at the lower strength (*p* < 0.0125). The efficacy of Spiromax® was significantly decreased by inhalation volume at every tested volume compared with the 4 l volume; at an inhalation volume of 1 l and below, the decrease in DD was approximately 20–50% relative to that obtained at 4 l, depending on the dose strength (Figure 1).[31] This can have important implications for patients with obstructive lung diseases, as their inhalation volume capacity can often be below 1 l.[54] Other devices that are inhalation volume-dependent include the HandiHaler® (Boehringer Ingelheim) and Breezhaler® (Novartis).[48,44] Assessments with the Diskus® DPI showed that inspiratory volume capacity (IVC) when using the device was significantly higher in healthy volunteers than those with asthma, COPD or neuromuscular disease (all *p* = 0.004). The IVC for asthma, COPD and neuromuscular disease were 72%, 69% and 46% of those in the healthy volunteer group, respectively.[35]

## Comparative clinical efficacy of different devices as drug delivery systems

### *The varying effectiveness of medication delivery to the lungs*

Comparative clinical efficacy between device types is difficult to summarise, for a number of reasons. Individual drugs or drug combinations can be compared between devices, but conclusions cannot be made in general terms. Any comparison will always depend on the particular formulation and device involved. Randomised controlled trials (RCTs) often do not take into account the confounding nature of the type of device used, or sometimes simply fail to name the device used at all.[16] RCTs may not fully predict the efficacy of a therapy in a clinical setting, due to their inclusion of idealised, homogenous patient populations, and because of real-life variations in inhaler technique and device characteristics that are difficult to replicate in trials. Furthermore, it is often a requirement in clinical studies for patients to be proficient with the device, and technique and adherence can be monitored closely by the study investigators.[83]

Real-world, retrospective database studies are, therefore, a more informative method of gaining insights into the comparative efficacy of devices.[83] A retrospective observational study of the UK General Practice Research Database evaluated the relative effect on asthma control of ICS delivered by pMDI, BA-pMDI and DPI.[83] For patients receiving a first prescription for ICS, the adjusted odds ratios (OR) for asthma control (as defined by no asthma-related hospital attendances, prescriptions of oral corticosteroids, or antibiotics for lower respiratory tract infections) were slightly better with BA-pMDI and DPI than pMDI (OR 1.08 and 1.13, respectively; *p* = 0.013). The adjusted rate ratio (RR) for severe exacerbations was lower for DPIs *versus* pMDIs (RR 0.88). For patients stepping up their ICS dose, asthma control was significantly greater with BA-pMDIs than pMDIs (OR 1.21; *p* < 0.001), but the DPI cohort showed no significant difference compared with pMDIs (OR 1.13). Rates of severe exacerbations were significantly lower with both BA-pMDI and DPI than with pMDI (RR 0.83 and 0.85, respectively; *p* < 0.001).[83] However, a retrospective UK database study in COPD patients supported the use of pMDIs over DPIs for the delivery of a fluticasone propionate/salmeterol (FP/SAL) formulation at two doses of FP.[84] FP/SAL 500/50 μg/day delivered *via* pMDI reduced

the frequency of moderate/severe exacerbations compared with delivery *via* DPI ($p = 0.032$), although there was no difference observed at a dose of FP of 1000 µg/day. The same study group had also previously shown the positive benefits of pMDIs over DPIs for achieving asthma control with FP/SAL in a retrospective UK database study.[85] A retrospective and multicentre Spanish study of fixed-dose ICS and long-acting β₂-agonist (LABA) combinations also showed that DPIs were associated with lower adherence compared with pMDIs in COPD patients, after adjusting for confounding factors ($p = 0.002$).[86] A real-world, historical, matched cohort study using data from two large UK databases examined the effectiveness of adding a spacer to a pMDI.[87] The study found no evidence that ICS administration by pMDI with a spacer was associated with improved clinical outcomes compared with a pMDI alone in patients with asthma. There were no significant differences in severe exacerbation rates between the two study arms for either fine-particle or extra-fine-particle preparations.[87] In contrast, a recent study showed that the use of an AeroChamber Plus® Flow-Vu® spacer (Trudell Medical International) with the Symbicort® (budesonide/formoterol) pMDI (AstraZeneca) in healthy volunteers increased total systemic and lung bioavailability of the drug relative to pMDI alone.[88] In subjects with poor inhalation technique, the use of a spacer increased the bioavailability equivalent to that seen in subjects with good inhalation technique without a spacer. An associated *in vitro* study found that the fine-particle dose was unaffected by the use of the spacer.[88]

A systematic review compared the results of 30 RCTs involving a range of inhalers in COPD and asthma patients for their clinical benefits, but found no demonstrable improvements in clinical outcomes between the devices.[16] Of note, in the comparison of tiotropium delivery *via* Respimat® Soft Mist™ Inhaler and HandiHaler®, equal clinical results were observed with a dose of 5 µg once-daily with Respimat® compared with 18 µg once-daily for HandiHaler®. These results were supported by a systematic review specifically comparing the efficacy and safety of tiotropium delivered *via* Respimat® or HandiHaler® in patients with COPD. The devices were found to be similar at doses of 5 µg and 18 µg (DD, 10 µg[48]), respectively.[89] In addition, no difference in the risk of death or cardiac adverse events between tiotropium Respimat® 2.5 µg or 5 µg and tiotropium

HandiHaler® 18 µg was found in a large randomised controlled trial.[90] Comparative RCTs and *in vitro* studies of budesonide/formoterol administered *via* the Spiromax® and Turbuhaler® DPIs have yielded mixed results. A randomised, double-blind, double-dummy efficacy and safety study in patients with persistent asthma demonstrated noninferiority between the two devices in improvement in daily trough morning peak expiratory flow (PEF).[91] In the biphasic Easy Low Instruction Over Time (ELIOT) study, Spiromax® was associated with a significantly greater proportion of patients achieving device mastery after initial training (94% *versus* 87%; $p < 0.001$), but the proportion of patients maintaining device mastery after 12 weeks was similar (59% *versus* 53%; $p = 0.316$).[92] In a budget impact model based on the results of the ELIOT study and an observational study which included Turbuhaler®,[93] Spiromax® was associated with potential savings in unscheduled healthcare costs compared with Turbuhaler®.[94] However, the model made a number of assumptions around the relationship between healthcare utilisation and inhaler errors. In an *in vitro* study, Spiromax®, which, from a patient perspective, visually resembles a pMDI but, in contrast to a pMDI, contains specific instructions not to shake the device in the patient information leaflet, was also significantly more affected by pre-inhalation shaking of the device than Turbuhaler®.[31] There was no difference in DD of budesonide between shaking and no shaking for budesonide/formoterol Turbuhaler® 160/4.5 µg, and only a small decrease when the 320/9 µg strength inhaler was shaken. With Spiromax®, DD decreased by 80% when the 320/9 µg strength device was shaken (Figure 2).

In summary, the results from RCTs often show little difference in the clinical effectiveness of inhalers when used correctly. The results from observational studies are much more inconsistent, due to the numerous real-world confounding factors that can adversely influence inhaler technique.[16,83]

## Device use in practice

### Definition of critical errors

It is often difficult to define exactly which device handling errors can be categorised as 'critical'. A broad definition favoured by the Inhaler Steering Committee is 'when a patient performs an error, displays imperfect technique or lacks knowledge

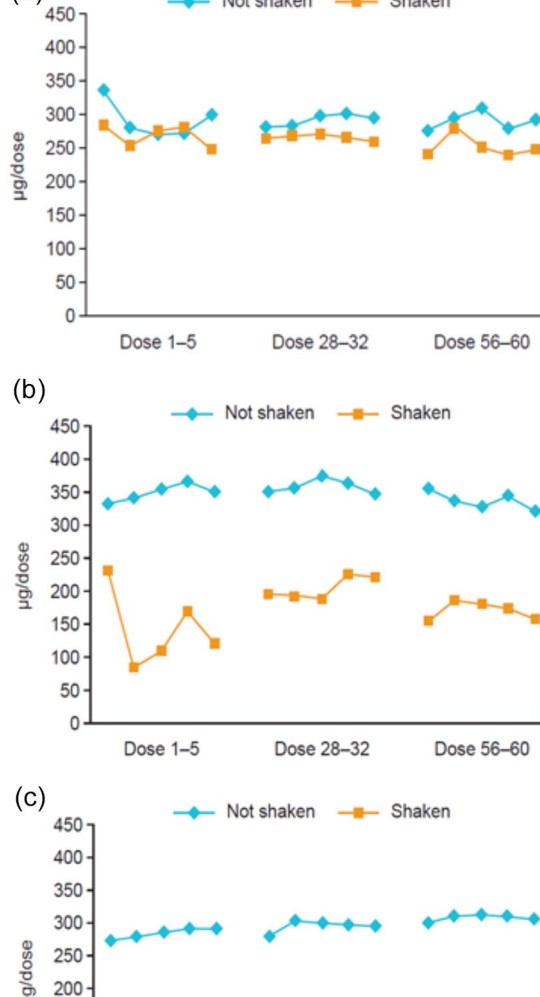

**Figure 2.** Mean delivered budesonide dose for (a) Turbuhaler®, (b) Easyhaler® and (c) Spiromax® when the device was shaken and not shaken.[31] For each combination device of budesonide/formoterol, the expected delivered dose of budesonide was 320 μg.

until an error has been scientifically determined as critical to the function or usage of a device in a real-world study, any such definition should be considered as purely theoretical.

*Incidence of inhaler technique errors*
The use of different study designs and inhaler technique checklists makes it difficult to accurately compare error rates between, or within, inhaler device types. The CRITIKAL (CRITical Inhaler mistaKes and Asthma controL) study identified critical errors as those that occurred frequently and were associated with poor disease outcomes.[97] The study included two DPI cohorts (Turbuhaler® and Diskus®) and one pMDI cohort [Evohaler® (GlaxoSmithKline)]. A common error in the DPI cohorts was insufficient speed and force in the inhalation (32% error rate for Turbuhaler®, 38% for Diskus®), while the inspiratory effort was not slow and deep enough in 47% of patients using Evohaler®. Other errors common to all devices were incorrect tilting of the head during inhalation and not exhaling before inhalation.[97]

*Common errors*
Several systematic reviews have attempted to quantify the most common types of critical handling errors. A qualitative review of 72 studies found an estimated overall error frequency of 87% for pMDIs compared with 61% for DPIs, although there was a high level of heterogeneity between studies for both device types.[98] The pooled estimate for critical errors with pMDIs was 46%, compared with 28% for DPIs. However, another systematic review assessing 38 studies in both asthma and COPD patients found that it was not possible to draw conclusions regarding the failure or misuse rates of a range of pMDIs and DPIs, due to the differing definitions, patient populations and assessment methods included.[99] A review evaluating 40 years of pMDI and DPI usage data concluded that incorrect inhaler technique is frequent and has not improved over that time period.[100] The review used an author-developed framework to enable comparison of a number of studies using differing devices in various patient populations over multiple time periods. Poor inhaler technique was observed in 31% of patients, although pMDIs had a higher average frequency of errors than DPIs. The addition of holding chambers to pMDIs did not have a substantial impact on error reduction.

on usage or maintenance of the inhaler device that is likely to significantly impair the delivery of adequate medication on all occasions'.[95] A systematic review of 123 publications found 299 different descriptions of critical errors, with multiple terminologies for exactly the same inhaler error between studies.[96] The most common definition was 'an action affecting the lung deposition of inhaled drug, resulting in little or no medicine being inhaled or reaching the lungs'. However,

Numerous other studies and reviews have evaluated error rates with various devices. An observational study of COPD and bronchial asthma patients in India found that at least one error was made during inhaler use by 94% of those using pMDIs, 82% using DPIs and 78% using a pMDI with a spacer.[101] In a Brazilian observational study, at least one error was demonstrated by 72% of pMDI users and 51% of DPI users.[102] A prospective cross-sectional assessment of COPD patient compliance with device technique instructions for pMDIs, pMDIs with spacers and DPIs showed similar results: three-quarters of patients performed at least one step incorrectly, with DPIs having the lowest percentages of incorrect inhalation technique.[103] Errors associated with the different device types identified from various other studies are shown in Table 2.

### Device-specific errors and critical errors

As well as differences in errors between device types, specific errors *within* device types have also been extensively studied (Table 4). In a study of 180 COPD outpatients in Turkey, correct use of pMDI, Diskus®, HandiHaler® and Turbuhaler® DPIs, as measured by the rate of performing all steps of application without error, did not significantly differ ($p = 0.082$).[104] In another observational study of almost 3000 COPD patients, over 60% could not perform a perfect inhalation in line with the device label, regardless of device type. Device-specific critical errors accounted for 15% of handling errors for Breezhaler®, 21% for Diskus®, 29% for HandiHaler®, 32% for Turbuhaler®, 44% for pMDI and 47% with Respimat®.[105] Failure to form a proper seal around the mouthpiece was found to be a common error with both Turbuhaler® and Diskus® DPIs in one systematic review.[98] In addition, a study of Jordanian and Australian asthma patients using the Diskus® DPI identified failing to exhale to residual volume prior to inhalation, not exhaling away from the mouthpiece, and inadequate postinhalation breath-hold as the most common errors with this device. The most frequent incorrect step with Turbuhaler® was not keeping the inhaler upright while loading the device.[106] An open-label study of device-naïve patients reported that the most common initial errors with Turbuhaler® were in the preparation of the device, whereas the inhalation stage was most problematic with Diskus®.[107] These results for Diskus® were supported by a Chinese study in

COPD patients that found that fast and extremely forceful inhalation was the most common error (94%), along with inadequate breath-hold after inhalation (90%).[108] In a real-world study, frequent inhalation errors with Turbuhaler® were the inability to exhale gently away from the mouthpiece without blowing into the device, and not satisfactorily completing the postinhalation breath-hold.[109]

As some patients may use their devices with minimal training, it is also important to consider error rates in inhaler-naïve subjects. In one study assessing how easily such participants could master their Spiromax®, Turbuhaler® or Easyhaler® device in a short space of time, a significantly larger proportion of participants were able to use Spiromax® without error ($p < 0.001$).[113] A common Spiromax® error was not holding the device upright, and Easyhaler® users often did not properly shake the device and hold the plunger down correctly. Participants using Turbuhaler® had problems with the preparation of the inhaler.[113] A comparison between the Ellipta® (GlaxoSmithKline) and Breezhaler® DPIs, again conducted among inhaler-naïve participants, reported fewer device handling errors with Ellipta® at both first attempted use (11% *versus* 68%, respectively) and after nonverbal device demonstration (2% *versus* 33%).[119] The most frequent difficulty that participants encountered with the Breezhaler® involved incorrect activation of the side buttons whilst preparing the device. Another study compared the error rates of Ellipta® and other common devices made by COPD and asthma patients after reading the patient information leaflet (PIL).[110] In COPD patients, there were significantly fewer initial critical and overall errors with Ellipta® than Diskus®, pMDI, Turbuhaler®, HandiHaler® and Breezhaler® (all $p < 0.001$). The most common critical errors were exhaling directly into the Ellipta® mouthpiece, not pushing the Diskus® lever back completely, poor actuation-inhalation coordination with the pMDI, and not twisting the base properly with Turbuhaler®. For both HandiHaler® and Breezhaler®, the most common critical error was the capsule not rattling, indicating that the dose was not received. In asthma patients, Ellipta® was not significantly different than Diskus® or pMDI for critical ($p = 0.221$ and 0.074, respectively) or overall error rates ($p = 0.186$ and 0.217), but there were significantly fewer critical errors *versus* Turbuhaler® ($p < 0.001$). The most common errors were the same as in the COPD

**Table 4.** Common inhaler errors associated with specific devices (general errors in Table 2).

| Device type | Brand | Common errors |
|---|---|---|
| DPI | Diskus®[97,105,107,110–112] | Reduced dose after preparation due to holding downward<br>Shaking the device after dose preparation<br>Lever not pushed back completely<br>Not closing the device<br>Not checking counter has decremented after inhalation |
| | Turbuhaler®[92,97,105,107,110–113] | Device not held upright when priming the device<br>Base not twisted until it clicks |
| | HandiHaler®[105,110,111] | Failure to insert or remove capsule<br>Not fully closing device capsule chamber<br>Not piercing the capsule<br>Capsule did not rattle<br>Failure to press and release button<br>Opening next blister when taking the capsule<br>Not checking if powder left inside capsule chamber/no second inhalation |
| | Breezhaler®<br>(Neohaler®)[105,110] | Failure to insert or remove capsule<br>Placing a capsule directly into the mouthpiece<br>Not fully closing device capsule chamber<br>Not piercing the capsule and failing to release piercing buttons fully before inhalation<br>Capsule did not rattle<br>Failure to press and release buttons<br>Pressing the two side buttons more than once<br>Not opening the inhaler to loosen capsule when whirring noise absent<br>When checking a stuck capsule, not loosening the capsule by tapping the base of the inhaler<br>Not checking if powder left inside capsule chamber/no second inhalation<br>Removing the empty capsule by touching capsule in the inhaler |
| | Genuair® (Pressair®)[114,115] | Not pressing the button all the way down before inhalation<br>Not releasing the button before inhalation, or pressing the button during or after inhalation<br>Inhaling when the control window is still red<br>Stopping inhalation when a 'click' sound is heard<br>Not repeating the inhalation even when control window is green<br>Not replacing the protective cap |
| | Ellipta®[110,111–114] | No 'click' sound after sliding the cover open<br>Shaking device upside down after a dose preparation<br>Not closing the cover |
| | Aerolizer®[116] | Not perforating the capsule once and releasing the lateral trigger<br>Not checking if powder left inside capsule chamber/no second inhalation |
| | Easyhaler®[112,113] | Failure to shake the device<br>Device not held upright when priming the device<br>Holding the plunger down when inhaling<br>Not emptying the device when loading more than one dose |
| | Spiromax®[92,113] | No 'click' sound after opening the cap<br>Inhaler not held upright when a dose is prepared<br>Not holding device upright with mouthpiece down<br>Shaking before or after dose preparation<br>Not closing cap postinhalation and loading a new dose |
| | NEXThaler®[117] | Not holding device correctly during loading<br>Not inhaling as rapidly and forcefully as possible (to overcome breath trigger threshold value) |
| SMI | Respimat®[105,116,118] | Lack of cartridge in the device<br>Incorrect assembly of the inhaler<br>Failure to twist the base<br>Not pressing down the inhaler<br>Incorrectly pressing for the number of puffs specified<br>Problems with co-ordination of actuation and inhalation (late start of inhalation causes exhaled dose due to the slow mist generation)<br>Inspiratory effort too high in combination with a low resistance, leading to too fast an inhalation |

DPI, dry powder inhaler; SMI, soft mist inhaler.

population.[110] In a study of patients with no previous experience of using a DPI, but who were allowed to read the PIL, the most common errors observed with NEXThaler® (Chiesi Farmaceutici SpA, Parma, Italy) and Diskus® were not inhaling rapidly and forcefully, whereas incorrect handling of the device for loading was most frequent for Turbuhaler®.[117]

### Storage

The importance of correct device storage has been highlighted in several studies, predominantly focusing on DPIs, which can be sensitive to humidity due to the fact that the dispersion of fine particles is impaired by moisture.[120,121] A study by Norderud Lærum and colleagues surveyed the storage conditions in which 738 Nordic asthma patients kept their devices, and found that 63% kept their maintenance inhaler in humid locations (including 42% storing them in the bathroom).[122] The kitchen is another common place to store medications, and is also associated with high temperatures and relative humidity (RH).[123] The impact of humidity on common DPIs was reported in a 3- and 6-month *in vitro* study with Turbuhaler®, Novolizer® (Meda Pharmaceuticals), Easyhaler® and Spiromax®.[120] For inhalers containing budesonide only, there was no significant reduction in DD or FPD with Turbuhaler® at ambient temperature/75% RH, whereas the DD of Novolizer® was significantly reduced at 6 months ($p = 0.01$). The Easyhaler® FPD was significantly reduced by 39% compared with baseline after 1.5 months and by 54% at 6 months (both $p < 0.01$), with DD significantly lower after 6 months ($p < 0.01$). With the fixed-dose budesonide/formoterol combination inhalers, budesonide FPD was significantly decreased after 1.5 and 3 months at ambient temperature/75% RH with Easyhaler® (both $p < 0.01$) and Spiromax® ($p < 0.01$ and $p = 0.02$, respectively), but not with Turbuhaler®. For the formoterol component, there was a decrease of 20% in FPD with Easyhaler® at 3 months but no change with Spiromax® or Turbuhaler®. The study showed that the susceptibility of devices to humidity differs, and the pronounced sensitivity to humidity of Easyhaler® was determined to have resulted in a clinically relevant decrease in FPD.[120] However, it should be noted that, conversely, another *in vitro* study found that moisture had no effect on DD and FPD with the Easyhaler® at 30°C/75% RH storage condition for 4 days.[124] The positive attributes of the Turbuhaler® after storage

in humid conditions detailed above have been ascribed to the tight cover and desiccant (within the device), which offer moisture protection.[120,125]

### Shelf-life

All DPIs are sensitive to humidity, and most inhalers must be stored in dry conditions below 25–30°C. Currently available DPIs have varying expiry dates, both in terms of unopened shelf-life and time-in-use once opened, depending on the respective humidity protection. The unopened shelf-life of most devices is 2–3 years, but the time-in-use once opened can range from 6 weeks (Trelegy® Ellipta®, GlaxoSmithKline) to 3 years (Symbicort® Turbuhaler®).[42,122,126] By comparison, Spiriva® Respimat® has an unopened shelf-life of 3 years and an in-use shelf-life of 3 months.[48] These shelf-lives are valid only if the devices are stored according to the manufacturer's recommendations, and the optimal drug delivery and FPD from first to last dose can only be guaranteed if the device is stored in the specific conditions and below a certain temperature. Failure to comply with these recommendations may have an impact on the correct functioning of the device and, in turn, on clinical efficacy. There is a general lack of data on this subject, although one study has shown that 63% of patients check inhaler expiry date less than monthly or not at all, and 30% of patients occasionally or frequently use their device after the expiry date.[122]

## Patient knowledge, perceptions and behaviours

As well as the multiple technical factors that healthcare providers must consider when choosing a device, there are also a range of human factors that can affect the suitability of a particular inhaler. Patient acceptance of a particular device should be taken into account, but there is limited evidence that this can improve disease control.[121] It has also been shown that patients frequently make errors such as storing their devices in suboptimal conditions, concurrently using more than one maintenance inhaler and exceeding the stated shelf-life, and ignoring, or are not being aware of, expiry date information.[122] Patients can often experience handling problems with even the most basic of common devices, and this poor level of knowledge and technique is associated with poor outcomes.[7,121] For instance, when using pMDIs, patients cannot

accurately determine when the device is empty without a dosage counter. This may result in patients putting themselves at risk by continuing to use an empty inhaler or, conversely, renewing the prescription earlier than is necessary.[121]

Factors associated with poor inhaler technique include age, disease severity, level and method of training, patient difficulties with instruction and polypharmacy.[7,15,93,127] Other factors, such as sex and education have also been shown to be important.[93,128,129] An observational study of the prevalence of inhaler mishandling in experienced patients found that critical mistakes with both pMDIs and DPIs were widely distributed.[93] The factors most strongly associated with inhaler misuse, independent of device type, were older age ($p = 0.008$), lower education level ($p = 0.001$) and lack of instruction from a healthcare professional ($p < 0.001$). There was also an association between perceived lack of medication efficacy and device misuse ($p = 0.015$), and a nonsignificant trend for reduced risk of inhaler misuse by women ($p = 0.064$).[93] The results of this study were supported by a Brazilian observational study which found that advanced age ($\geqslant 60$ years), low level of education ($\leqslant 8$ years of schooling) and a lower socioeconomic status increased the risk of device errors.[102] An historical cross-sectional study assessed the factors affecting DPI device handling errors in asthma patients.[129] Factors significantly associated with serious errors, defined as those potentially limiting drug uptake to the lungs, were female sex ($p = 0.032$), obesity ($p = 0.036$), lack of a university degree ($p = 0.006$), asthma-related hospitalisations ($p = 0.008$) and having poor asthma control in the prior 4 weeks ($p = 0.012$).

### Interventions to improve inhaler technique
Interventions to improve patient inhaler technique and adherence were evaluated in a Cochrane review of 29 studies in asthma. There were positive results for face-to-face and multimedia inhaler training on technique, both immediately after training and at follow-up.[130] Feedback devices also enhanced inhaler technique, and it was suggested that interventions providing inhaler training may bring some benefit to quality of life and asthma control. Analyses that used correct or 'good enough' technique as an outcome were generally deemed more useful than those that used a checklist score. Due to the differences in interventions, study populations and outcome measures, however, most of the evidence was considered to be of low quality and drawing firm conclusions was difficult.

In the historical cross-sectional study of DPIs referenced earlier, the absence of an inhaler technique review in the prior year was associated with making at least one serious error ($p = 0.012$).[129] The importance of such a review was also shown in asthma and COPD patients, where patients checked at least once at follow up had a lower risk of critical errors for both pMDIs and DPIs ($p = 0.0001$).[93] The effect of face-to-face training on these factors was shown in a study of asthma and COPD patients.[128] Before training, male sex, higher level of education, living in city *versus* rural locations, longer duration of disease, specialist follow up and more frequent hospitalisation were all associated with correct inhaler usage. After training, none of these differences was significant. There was less improvement in the correct use of the device following training in the pMDI group than the DPI group.[128] The effect of training was also apparent in a study showing that the majority of participants made errors when using Spiromax®, Turbuhaler® and Easyhaler® intuitively. After instruction by a healthcare professional, all devices had been mastered by >95% of participants.[113] A further study has suggested that a clear and easy-to-read PIL can help patients' inhalation technique, although training limited to reading the PIL only is not recommended and previous studies have associated this with frequent errors.[117]

Inhaler technique reminder labels have been shown to improve retention of correct inhaler technique in DPIs, when used in conjunction with training.[131] In a randomised, active-controlled study, patient inhaler technique was assessed and correct technique was then demonstrated to the patient until mastered. Any incorrect steps in the initial assessment were highlighted on the label affixed to the inhaler. The reminder labels resulted in significantly less decline in inhaler technique scores after 3 months for both Diskus® ($p = 0.022$) and Turbuhaler® ($p = 0.003$).[131]

### Problems associated with using multiple devices
The use of multiple device types has been shown to have a negative association with correct handling technique and patient adherence to therapy, and it is recommended to restrict patients' usual inhaled medication to as few

devices as possible.[121,132,133] A real-world observational study of COPD patients compared those using an additional inhaler device requiring a similar inhalation technique to their existing device(s) with those prescribed a device with a different technique. Each cohort included >8000 patients, and the 'similar-devices cohort' showed a lower rate of moderate/severe exacerbations (incidence rate ratio 0.82) and were less likely to be in a higher-dose short-acting $\beta_2$-agonist (SABA) group than the 'mixed-devices cohort' (adjusted proportional odds ratio 0.54).[132] The results of this study corroborated the findings of an earlier retrospective, observational study that found that asthma patients prescribed the same type of BA-pMDI for ICS controller and SABA reliever therapy were significantly more likely to achieve asthma control and have fewer severe exacerbations than patients using a BA-pMDI controller and a separate pMDI reliever.[133]

### Patient-related problems associated with particular devices

Specific errors due to lack of, ignorance of, or nonretention of, training can be an important factor in the correct functioning of a device. For instance, before using a suspension pMDI, the device should be shaken to prevent sedimentation of the drug, but advice on how and when to do this can differ between devices.[121,134] One study investigated the effect of a delay between shaking and firing four suspension pMDIs and one solution pMDI.[134] The devices were shaken for 5 s before being actuated at various time delays. The amount of drug delivered from the solution pMDI (QVAR® 100 Inhaler; Teva UK Ltd., Castleford, UK) was consistent across all shake-fire delay times tested,[134] as shaking is unnecessary for this formulation.[121] The three suspension pMDIs had increasing drug delivery and one had decreasing drug delivery with increasing delay time. The mass of drug delivered by the Ventolin® Evohaler®, Flovent® HFA (GlaxoSmithKline, Research Triangle Park, NC, USA), and Airomir® Inhaler (Teva Pharmaceutical Industries, Espoo, Finland) after a 60-s shake-fire delay was 346%, 320% and 230% of that with a 0-s shake-fire delay, respectively. For the budesonide/formoterol pMDI (Symbicort®, AstraZeneca, Cambridge, UK), the delivery of both the budesonide and formoterol components was reduced with 20–60 s time

delays. With a 60-s delay, the DD was approximately 75% of that with a 0-s delay time. The authors concluded that specific guidance should be given on the timing of actuation after shaking a pMDI.[134]

Breath-hold time is a step that patients using both pMDIs and DPIs can experience problems with.[102,129] Users of both pMDIs and DPIs are recommended to hold their breath for at least 5–10 s after inhalation to enable optimal drug deposition in the lungs.[135,136] A study of the inhalation of salbutamol *via* pMDI and valved holding chamber in children with asthma found no improvement in peak expiratory flow between single maximal inhalation with breath-hold and five tidal breaths.[137] Other studies have supported common advice given to patients for DPIs, such as exhaling away from the inhaler immediately prior to inhalation,[138] and holding the device in the correct position; a finding especially significant for Breezhaler®.[139]

### Recommendations for clinical practice

The initial selection of a device should include a number of patient-related considerations. The prescribing physician should assess factors such as a patient's age and the peak inspiratory flow and inhalation volume they are able to generate.[15,31,35] A patient's sex and education level should also be taken into account, as they have been shown to potentially contribute to inhaler misuse.[93,128,129] For patients with physical or cognitive impairment, a device with fewer or more simple preparation steps or requiring less complicated inhalation manoeuvers should be chosen.[12] Once-daily dosing may be an advantage for these patients. Conversely, with twice-daily dosing, many patients appreciate the reassurance of inhaling their second dose before going to bed to decrease the likelihood of night-time symptoms.

Assessment of the patient's overall status can then be used to inform the decision on the most appropriate type of device. The physician should familiarise themselves with the various device characteristics that can affect drug delivery, and consider how this may impact on their suitability. Factors to review include the influence of inhalation flow rate and inhalation volume. For patients with poor lung function, certain DPIs may not be the most appropriate choice, due to their varying

internal resistance to air flow.[6,31] Common errors and problems associated with particular device types should be reviewed (Table 2), particularly with regards to the inhalation technique requirements and the number of steps needed for preparation and usage. Availability and cost issues will naturally form part of this decision, but should not be prioritised at the expense of the aforementioned considerations.

Once a device type has been decided upon, the inhalation profile and common errors associated with the individual devices should then be assessed in more detail. Comparative clinical data between devices from RCTs is often conflicting, although results from database studies can be more informative. *In vitro* studies can also be useful for analysing technical aspects of inhaler devices, although the clinical relevance of these findings is not always clear. The various devices are all associated with different common errors (Table 4), and their relative importance will vary depending on the individual patient. This is also true of flow rate-dependency, the effect of inhalation volume, and susceptibility to temperature and humidity.[31,32,120] A personalised approach should, therefore, be adopted when assessing these factors. The use of extra-fine particle formulations can be considered, although clinical data are mixed.[26] The use of multiple devices should be limited, where possible;[132] if this is unavoidable, then similar device types should be used to minimise the risk of errors.

Once an inhaler has been selected, it is important to ensure good inhaler technique is mastered and then maintained over time; thus, a number of factors relating to training have to be considered. Training, especially face-to-face training and regular inhaler technique reviews, has been shown to be effective,[93,130] and physicians should be aware that more frequent or in-depth training may be required for older patients. When commencing treatment with a new device, even with devices that have low error rates when used intuitively, face-to-face training with a trained professional is recommended.[110] Training should be tailored to both the patient and to the particular chosen device, focusing on the steps where errors are most likely to occur, such as in the preparation of the device, correct positioning and the particular inhalation technique required (Table 4). Patients should also be provided with clear practical guidance about devices, such as, for example, the

deleterious effect of unsuitable storage on DPIs and the varying effects of shaking and shake-fire delays with pMDIs.

## Future developments, new technologies and research gaps

### New technologies

Pharmaceutical companies are continuously trying to innovate and improve on the existing inhalation technologies available. The incorporation of modern technology into inhaler devices is chiefly aimed at improving drug delivery, reducing device errors, improving patient adherence and monitoring and managing patients' disease states.[140–142]

Previous advances in capsule-based DPI systems have mainly focused on incorporating lower internal airflow resistance, and the utilisation of feedback mechanisms such as audible and visual cues. The Breezhaler®, for example, emits a sound as the powder is inhaled, and the clear capsule allows patients to see that the powder has been fully inhaled, whilst also indicating when the device is empty.[143] New cosuspension technology uses low-density phospholipid particles to suspend micronised drug crystals in an HFA propellant, meaning multiple drugs can be administered *via* a single pMDI in a uniform manner.[140,141] The low-density phospholipid particles increase the physiochemical stability of the drugs and can also reduce the effects of a shake-fire delay. *In vitro* tests have shown highly reproducible and consistent drug delivery,[144] and a study in healthy volunteers has shown effective lung deposition.[145] An *in vitro* study of a fixed-dose LAMA/LABA combination administered by the cosuspension pMDI, Aerosphere® (AstraZeneca), achieved reproducible dose delivery and an FPF $> 55\%$. This was maintained across variations in flow rate, and drug delivery was constant under conditions of simulated patient handling errors, such as variable shake technique and delays between shaking and actuation.[141]

The first in-built inhaler monitoring technology was developed in the 1980s, mainly to assess adherence to medication, and this has evolved over the years to incorporate various other sensing functionalities.[142] Development of the Smart Inhaler Tracker (Adherium) to store the dates and times of inhaler actuations led to the development of more sophisticated devices compatible with most

common inhaler types, for example, SmartTurbo®, SmartDisk® and SmartTrack® (Adherium).[142] The Propeller Health device was the first to incorporate Global Positioning System (GPS) functionality, in order to map potential triggers of exacerbations. Newer devices, such as Care TRx® (Teva Pharmaceutical Industries), Sensohaler® (Sagentia), Inspiromatic® (OPKO Health) and the T-Haler® (Cambridge Consultants) incorporate functions capable of monitoring parameters as diverse as PEF, inhalation flow and volumetric flow rate.[142] The development of wearable biosensors, in conjunction with smartphone apps, can also be used to monitor a wide range of physiological parameters.[146]

The incorporation of dose-memory and dose-reminder functions in inhalers can have a positive effect on adherence and can increase confidence in self-management behavior.[147,148] In the 12-month STAAR study in children with asthma, for example, clinical review of electronic adherence monitoring data and dose reminders were shown to improve average adherence, and reduce the number of courses of oral steroids and hospital admissions compared with nonreview and no reminder function.[149] In a randomised controlled trial in children with asthma, an electronic monitoring device with an audiovisual reminder function led to significant improvements in adherence to ICS maintenance inhalers.[150] Adolescents, in particular, appear receptive to smartphone apps with reminder functions to facilitate adherence.[151] The recent development of smart inhaler technology such as that incorporated in the Bluetooth-enabled Turbu+™ device (AstraZeneca/Adherium), which monitors actuations and provides feedback and reminders to patients through a mobile app, and the Connected Inhaler System® being trialled by GlaxoSmithKline/Propeller Health (ClinicalTrials.gov identifier: NCT03380429) have built on these study results.[152,153] Recent experience with Turbu+™ in Italy, for example, exemplified improved adherence.[154]

Digital health developments have also shown great utility in the management of device errors, and are now able to provide detailed feedback on patients' device competence.[142] The SmartMist™ (Aradigm) and MDILog™ (Westmed Technologies) have both included sensing capabilities to facilitate the assessment of inhalation technique. The MDILog™, which is widely used in clinical research, is designed to attach to the plastic casing of standard inhalers. The device includes an inhaler actuation sensor, as well as an accelerometer for the detection of inhaler shaking and a sensitive temperature sensor for the assessment of inhalation. Inhalation detection technologies can be used to coach patients on correct device technique, and further developments include wearable biosensors and smartphone apps that have the potential to correct inhaler errors with, for example, pop-up instructional videos based on real-time measurements.[146] Technology can also be integrated into spacers to monitor whether a pMDI has been shaken, the shake-fire delay, and the volume of inhalation from the device.[155] Feedback can then be provided to the patient on whether or not the correct technique has been used. This kind of technology, along with other innovative e-health developments, such as mobile communication technology (mHealth), electronic reminders, telemedicine and inhaler tracker interventions, have the potential to reduce the resource burden on healthcare systems and provide optimal and personalised asthma management to patients.[156,157]

### Knowledge gaps and future research needs

The main data gaps in the inhaler landscape are as a result of the lack of head-to-head trials comparing clinical outcomes for the same drug (or combinations) delivered *via* different devices. Future research needs include the standardisation of study designs, patient populations and outcome measurements to enable robust comparison of such studies. Less stringent inclusion criteria in clinical trials may enable evaluation of devices in a setting more representative of that in the real world, and adherence information and 'good enough' technique should also be reported more frequently.[130] The continued integration of the latest technology for assessing inhaler technique is essential to optimise patient self-management, as this is currently only present in a small amount of available devices. Areas of focus should not only be on device usage, such as sensors to detect inhaler shaking, but also on physiological and environmental parameters that can all affect correct inhalation.[142] mHealth, such as the use of smartphone apps, biosensors and automated incentives to reward adherence is likely to play an increasing role in the management of chronic conditions such as asthma and COPD, however, much of this new technology still needs to be tested for its feasibility, acceptability to patients and sustainability.[146]

## Conclusion

While there are a multitude of studies evaluating the characteristics of the available inhalation devices, it is often harder to find practical considerations to guide healthcare providers on appropriate device selection. This review has shown that, although availability and cost considerations may limit the choice of inhaler, it is vital that the needs of the individual patient, as part of a personalised treatment approach, are the primary focus.

The comparative data between device types are mixed. The studies included in this review tended to find that pMDIs were associated with the most handling errors, although all device types were associated with incorrect technique to some extent. General errors common to both pMDIs and DPIs include lack of pre-inhalation expiration, maximal inhalation after expiration (DPIs) and no postinhalation breath-hold; errors that could be prioritised in inhaler technique reviews. Other main areas of difficulty with DPIs were preparation of the device, while problems with actuation-inhalation coordination and the speed and depth of inhalation were frequently seen with pMDIs. There are conflicting data on the relative merits of individual brands of inhalers. Each device is, however, associated with a particular set of common errors, therefore training should be individualised to specifically focus on potential errors associated with the device used. Where applicable, devices that incorporate the latest monitoring and patient feedback technology should be considered, to give the best chance of improving patient inhaler technique and adherence, and ultimately improving disease control.

## Acknowledgements

The authors would like to thank Stefan Courtney of inScience Communications, Springer Healthcare, UK, for providing medical writing support, which was funded by AstraZeneca in accordance with Good Publication Practice (GPP3) guidelines (http://www.ismpp.org/gpp3).

## Author contributions

All authors took part in the planning of the manuscript, discussion of the results of the literature search and the writing of the manuscript.

## Funding

The author(s) disclosed receipt of the following financial support for the research, authorship, and publication of this article: This review was supported by AstraZeneca.

## Conflict of interest statement

Federico Lavorini has received fees for speaking and grants for research from AstraZeneca, Boehringer Ingelheim, CIPLA, Chiesi, Menarini International, Novartis, Orion and Teva, and has participated in advisory boards arranged by Boehringer Ingelheim, GlaxoSmithKline, Menarini International, Orion and Trudell.

Christer Janson has received payments for educational activities from AstraZeneca, Boehringer Ingelheim, Chiesi, Novartis and Teva, and has participated in advisory boards arranged by AstraZeneca, Boehringer Ingelheim, Chiesi, GlaxoSmithKline, Novartis and Teva.

Fulvio Braido has received fees for speaking from AstraZeneca, Boehringer Ingelheim, GlaxoSmithKline, Chiesi, Menarini International, Novartis, Teva, Zambon, Dompè and Lallemand, and has participated in advisory boards arranged by Boehringer Ingelheim, GlaxoSmithKline, AstraZeneca, Novartis, Mundipharma, Teva and Chiesi.

Georgios Stratelis is a full-time employee of AstraZeneca.

Anders Løkke has received fees for lectures and for educational activities from AstraZeneca, Boehringer Ingelheim, Chiesi, Novartis, GlaxoSmithKline, Mundipharma and Novartis, and has participated in advisory boards arranged by AstraZeneca, Boehringer Ingelheim, Chiesi, GlaxoSmithKline, Pfizer and Novartis. Furthermore, he has received grants for research from Pfizer, Boehringer Ingelheim and Novartis, as well as research-related travel accommodation from AstraZeneca, Boehringer Ingelheim, Chiesi, Novartis, GlaxoSmithKline, Mundipharma, Orion and Novartis. Anders Løkke has previously been/is currently a principal investigator in pharmaceutical company-sponsored research studies for AstraZeneca, Boehringer Ingelheim, Chiesi, GlaxoSmithKline and Novartis.

## ORCID iD

Federico Lavorini [iD] https://orcid.org/0000-0002-3293-2123

## Supplemental material

The reviews of this paper are available via the supplemental material section.

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
