## [Author_response_to_reviewer_comments – Supplemental material for What to
consider before prescribing inhaled medications: a pragmatic approach for
evaluating the current inhaler landscape · Therapeutic Advances in Respiratory Disease]

Author response to reviewer comments:

Reviewer: 1

Is there an advantage of once daily dosing versus twice daily?

This is not a topic that was included in the initial scope of the manuscript; however, this consideration has now briefly been included in the 'recommendations for clinical practice' section.

What are the recommendations about breath holding? Is it similar for DPIs and pMDIs?

There is some information on this already included in the 'characteristics of available devices' section, but further explanatory text has been added in the 'patient-related problems associated with particular devices' section.

What is the recommended breathing pattern through a VHC?

The recommendations vary depending on the device. A sentence has been added to the characteristics of available devices' section to reflect this.

Reviewer: 2

Inclusion of pictures of various inhalers would enhance the review. Providing links to websites with the correct technique of use would also be helpful.

As the review is already fairly long and includes several tables and figures, we have not included pictures; however, links to useful websites with general information about inhaler devices have been included.

Reviewer: 3

It has been a pleasure to review this manuscript. It covers a large body of evidence and hence I would consider it a highly ambitious review. It covers a lot of topics/concept relevant to inhaler use and I commend the authors for undertaking it. However, as a result of its broad scope, I think it does suffer from a lack of direction and hence I think a more strategic and structure approach to its composition and flow is required. I have included some suggestions which I hope will be helpful.

At times I feel it is detailed, at times, incomplete.

At times it appears that the authors lack focus and stray from what their title suggests this review is about – a pragmatic approach and what to consider when prescribing.

Finally, what I feel it is lacking is an evidence-based series of recommendations, which is what is most needed in this field of research and practice. My question is, what will health care providers get out of this with regards to selecting devices? Will device selection now be more systematic? Will it be more individualised? Or will health care professionals simply be more informed? At this

stage it is a literature review of a series of topics, often a literature review of secondary sources (e.g. Table 1 and its reference to two review manuscripts).

There has been explosion of manuscripts on inhaler technique and devices over the last 5-7 years in particular. Further publications really need to add something more. This one does have the potential to do so but needs some reformatting and re-aligning.

We have incorporated a number of changes to the content and structure, based on both the comments above and the more specific comments below, which we hope will greatly improve the focus of this manuscript.

The objective of this literature review needs to be far more clearly articulated. The objective should be clarified with regards to 'choice' i.e. from whose perspective? The patient? The prescriber? Based on outcomes? It is not completely clear. Whatever the objective is, the reader should not be surprised when they read e.g. the section on page 4 which describes how the results are separate into categories – currently, there is no clear connection.

The objective is to consider the choice from a healthcare provider perspective, which has now been made clearer in the Introduction.

I suggest that the authors consider reformatting the review and perhaps reconsider the subheads e.g. rather than 'characteristic of available devices' use the heading "What is the difference between different devices". Also, instead of "Comparative clinical efficacy between devices" replace with "Are some devices more effective in delivering medications to the lungs than others?" This is just a suggestion as I don't think the headings as they currently stand work.

The manuscript has been reformatted to hopefully give a more streamlined flow to the text. The majority of headings and subheadings have been reworded so that they are more descriptive.

Also difficult to tell which are major headings and which are subheadings e.g. it appears that Characteristics of available devices is a heading under which pMDIs, spacers and valved holding chambers etc. and Comparative Clinical Efficacy between devices" all sit. I suggest a reformatting of the content. As an example perhaps something like the following:

- 1. Characteristics of available devices (under which I would suggest including technical features and their impact on use).**
- 2. Comparative clinical efficacy of different devices as drug delivery systems**
- 3. Device Use in practice (to include common errors, device-specific errors, 'critical' errors technical features which impact on ability to use correctly, most common errors, patient-driven errors, which include ability to perform inhaler manoeuvres etc.)**
- 4. Recommendations for practice (to include recommendations for initial selection of devices, recommendations for adding on devices, recommendations for improving inhaler use over time)**

5. Future developments, new technologies and research gaps.

The main headings have been numbered and the new sections have mostly been based on the above suggestions.

Some more specific comments below as they relate to specific sections:

Methodology

Please include the keywords/search terms.

The main literature search terms have been included in the main body of the manuscript to provide further clarity. However, due to the amount of search terms used, the bulk of them remain in the appendix.

Results

The importance of inhaler device choice – I think this should be in the introduction, especially if a narrative around expert opinion beyond evidence is a significant part of this section. There are so many concepts raised in this section which don't seem to add to the story but rather raise questions about what is important in device choice.

This has been moved to the Introduction, as suggested.

“The importance of Inhaler Choice”

I also think that the wording chosen needs to be more accurate to the data because in this field of research and practice, due to the difficulty in conducting high quality studies which answer specific questions about inhaler technique or ability to use, we have been far to ‘lose’ in the connections we draw from the available evidence to answer the questions we are really trying to answer. E.g. Page 5 “Patient acceptance and their ability of use certain devices is also important.....”. None of the references included in this statement i.e. references 14, 15 and 16 actually tested the patient’s ability to use. They all rely of patient-reported measures and they refer to adherence i.e. when the patient is likely to use but not their actual ability to use. Therefore, while this statement includes the words ‘may’ and ‘are likely’, it actually does not bring us any closer to the answer we are seeking.

This has been reworded to try and provide more clarity. However, this section is more of an introduction to the issues faced by healthcare providers when selecting an inhaled device, so it is not intended to provide any firm guidance on choice of device.

The reporting of the Delphi work is over-emphasised and open to misinterpretation by the reader. Regarding the data relating to the Delphi consensus statements, once again this needs to be interpreted with caution. Traditionally Delphi surveys are administered to cohorts of experts and hence, it is regarded as an effective method for reaching consensus amongst EXPERTS and in this case, with regards to factors that impact on them selecting devices. Now the consensus has come up with something quite logical, that choice is driven by whether the patient can use the inhaler, previous experience, ease of teaching the inhaler etc. However, when it comes using a device for the first time, how is the patient’s ability to use the device assessed? Is it trial and error? Is it what the prescriber perceives to be easy? And based on all that, can we then consider that some inhalers are actually easier to use? None of which were high recommendations of these Delphi panels???

Further to this, with regards to reference 16, with regards to ‘usability’ and ‘ease of training’, the particular reference is actually not able to determine this.

In addition to this, with regards to the experts’ views on costs, it is important to consider this with regards to the research of Laba et al., 2019 in JACI as it relates to the impact and importance of costs in relation to respiratory medication use by patients.

Therefore, I think this section on “The importance of inhaler choice” really should be about why choosing an inhaler is important and any actual evidence that proves this point – not what is currently included in this section. Alternatively, the heading for the section can be amended.

The wording of the Delphi study conclusions, as well as the other studies mentioned, has been revised to reflect the reviewer’s comments. Further explanatory text has been added to address the concerns about how these surveys were conducted, and the Laba et al. 2019 reference has been cited. The heading to this section has also been amended slightly to reflect these changes.

Characteristics of available devices

I think this should be in a table rather than text. If put in a table, I think it would be highly cited. This is not novel data but important references which are the basis of technology in this field.

Although we agree that this is a very good suggestion, a lot of data that was previously included in this section has already been tabulated during the drafting of the manuscript (Table 2). Therefore, we have left this section as it was.

Comparative clinical efficacy between devices

I would suggest that this section needs significant amending. This is a complex section, due to the challenges on actually getting data on the comparative efficacy of different devices therefore requires a more careful approach to presenting this to the reader. A more systematic approach to reporting the studies here needs to be taken. Either describe all the study designs or not. I actually think a table which includes study designs, comparison made and outcomes which summarise the results would be more helpful. The message is lost in all the text. There is a whole mix of comparisons, study designs, in vitro, clinical studies included here. A more systematic approach to reporting this is required.

In essence, there are actually only two comparisons that have been made, which can possibly give us insights into the difference in clinical efficacy related to device alone i.e. the comparison between the soft mist inhaler vs handihaler and the comparison between the Spiromax and the TH.

Finally, there is no conclusion to this section and I would suggest that the reader will need one.

Although we agree that a table would be useful, we have not created one for this section, as there are already a number of large tables included in the manuscript. However, we have attempted to improve the description of the included studies, and have included more information on study designs, where relevant. We have also restructured this section to improve the flow and make it more systematic. A short conclusion has also been added.

Below are some general points:

Please state in more detail why it is difficult to compare and summarise the difference between the clinical efficacy as it relates to devices alone. This is important for the non-expert reader. And

also, how a real life retrospective database study can be used to give us insights. I.e. why that specific design can be helpful here.

A discussion on the relative merits of RCTs and database studies has been added, which it is hoped will provide more context around the difficulties of comparing clinical efficacy between devices.

Please remove “Spiromax, which resembles a pMDI...”. It visually looks closer to a pMDI than anything else, but it should not be stated as being anything like a pMDI in an expert document like this. It does not technically resemble a pMDI and hence should not be stated as such in this manuscript. Perhaps one could say “For patients, the appearance of a Spiromax resembles the appearance of a pMDI hence in real-life studies, patients has tried to carry over some inhaler technique steps from the Spiromax to the pMDI, such as shaking the inhaler” I am certain the authors can selected better wording that I have suggested, but definitely need to reword the current text.

This has been reworded accordingly.

Device handling and storage errors

I would suggest that the following subheadings in the following order would work better for the content:

Incidence of inhaler technique errors,

Common errors

Device-specific errors

Critical errors

These subheadings have been used to divide the content of this section more appropriately.

Below are specific comments which I place in order in which the authors have presented the literature:

Definition of critical errors

It should be made crystal clear that the only way to determine which errors are critical in real life is if it has been tested in a scientific way i.e. as per the Critikal study. Everything else, every other definition, consensus definition etc. is just theory! It does not mean anything until it is proven with scientific fact.

A sentence to this effect has been added.

It should be stated that the Sanchis interpretation of “poor inhaler technique” is an author-developed framework to enable the comparison of multiple studies with multiple devices in multiple patient populations over multiple time periods over the last 40 years.

Explanation added, as requested.

“Numerous other studies and review have evaluated the effect of inhaler technique on asthma control.” This sentence seems out of place – this section seems to be on what errors are common?? What does this sentence then refer to? These studies should be included in the introduction relating to why inhaler technique is important not here?

'Asthma control' has been removed from the sentence. This sentence should now be in the correct context.

There are many more and much better references to use than the Portuguese study noted here (41). Why was this study signalled out?

On reflection, we agree that this study is perhaps not the most appropriate to include. We have replaced this with another more comprehensive study.

I think that Table 2 should be the table of reference for this section should it not? Table 3 related to Device-specific errors, which appears in the next section? I think that Table 2 in Sanchis et al., 2016 or Table 1 in Bosnic-Anticevich et al., 2018, which is modified from Sanchis is more useful than the table provided. Are all these listed in Table 2 actually 'common' or 'possible' or 'reported' – this of course matters.

The reference has been changed to Table 2, and Table 3 has been included as a reference for the next section. The table has been left as it is – it represents common errors that were reported in the referenced studies.

Device-specific errors

This section is difficult to follow. The reader just wants to know, which errors are specifically important for different inhalers. The authors include interventions and technique over time and effectiveness of different training is included – these are all distraction from what I think the reader will expect given the heading to this section. I think a good table with some explanation is all that is required here.

Further, there are so many studies that have published on the errors associated with different devices – what were the criteria for only selecting a handful here?

Specifically: "In a study of 180 COPD.....correct use of pMDI...did not significantly differ.." What is meant by 'did not significantly differ..', with regards to what? The number of errors, the nature of the errors? This needs to be clarified.

This section has been restructured. Some parts have been moved to the section on patient knowledge, perceptions and behaviours. The studies included were those that were assessed as being the most robust in the literature search results. The definition of 'significant' in the study mentioned above has been clarified.

Storage

If the studies related to FPD or DD then I think an explanation of these drug characteristics and humidity should be articulated up front.

"The difference in resistance to humidity.." why not just say "the impact of humidity on different DPIs....." The term "resistance" and DPIs has connotations that do not relate to humidity.

"..single budesonide inhalers.." are you referring to budesonide-only containing DPIs??

These studies are not explained well.

What is the conclusion then, which devices are affected by storage conditions and which are not?

A brief explanation of the effect of humidity on DPIs has been included. The suggested wording changes have been made, and we have attempted to make the study explanations more transparent. A short conclusion has also been added.

Shelf-life

What about the SMI? Does it not have an expiry date once opened?

Handihaler? Capsules?

The shelf-life of the Respimat SMI has been added. However, it is important to point out that this section was meant as an overview, rather than a systematic review of all devices. Therefore, we have not included data on HandiHaler or capsules. More detailed data regarding shelf-lives and expiry dates are included in Table 4.

Particle size considerations

Ok. So does particular size matter in real life and should this be a consideration when prescribing – what is the pragmatic approach here.

I think this section should be divided into theoretical impact of particle sizer and real-life studies on impact of particle size.

This section has been split into two sections, as suggested.

Inhalation flow rate

I like this section, however for the sake of being pragmatic, perhaps the recommendation by Seheult et al., that a spirometric PIFR of 196L/min should warrant further assessment of suitability of the device (at least the Diskus).

The explanation of reference 34 is not clear. It should also be noted that this study was in vitro. These are theoretical studies, when it comes to a pragmatic approach. This needs to be clear for the reader, because one could question the implications of in vitro results for a pragmatic approach – perhaps important but not discriminatory to decision-making??? Not sure? But certainly has to be noted.

The recommendations about spirometric PIF from Seheult et al. have been added, and the explanation of reference 34 (now reference 31) has been improved to include explicit mention that this study is in vitro.

Inhalation volume

Why is inhalation volume described as a patient-related factor but inhalation flow rate/PIF is not?

What is the relevance of reporting all the regulatory issues? Are you saying, what are the implications for the reader and their decision to prescribe a particular inhaler?

I think it would be interesting for the authors to provide a hypothesis regarding why these differences are seen between the different devices and doses of medication?

Reference to 'patient-related' has been removed. A short explanation of the importance of the regulatory issues discussed has been incorporated.

The Need for patient education

The heading, "The need for patient education" does not fit the content that follows. The need for patient education should be a fundamental recommendation. Perhaps this section should have the following heading: "Patient knowledge, perceptions and behaviours". I need to think more carefully about how the multi inhalers section fits into this? Maybe should be a separate subheading of its own?

This section has been restructured to include the heading and subheading suggested.

Patient perceptions and behaviours

Concluding that Levy et al., 2016, or in fact the primary reference by Choroa et al., 2014 concludes that patient preference makes a difference to outcomes is just not correct. This needs to be amended. Choroa et al., 2014 comes to the conclusion, as does Levy et al., 2016, that there is no conclusive evidence that patient satisfaction impacts on outcomes. The current text may mislead the reader – it needs to be amended.

Are the finding report from Norderud et al., 2016 related to behaviour or would we put this down to knowledge?? Heading should therefore include 'knowledge'.

"Factors associate with poor inhaler technique...." Would have been a nice statement to start this section with, rather than para 2??

What does 'serious' error mean when we know that the only error that can be considered critical is that proven by evidence (Price et al and Critikal).

The text has been amended with regards to Levy et al. and Choroa et al. We agree that "Factors associated with poor inhaler technique..." would be a good opening sentence; however, we have left it as it is because we feel this goes better with the flow of the next section. The definition for 'serious' error has been explained in the text.

Device-specific errors – do we need to have this heading again? Is there no more appropriate heading for this section? It seems to be more about education and lack of knowledge of certain steps – would that not be a better heading?

This subheading has been replaced with 'Patient-related problems associated with particular devices'.

New technologies

This is now focusing on adherence and to this point, while adherence has been noted by the authors as it relates a handful of manuscripts, it does not fit here – it needs an explanation and context.

Firstly there needs to be some comment about what the purpose is of new technologies, what problems are they trying to address and why trying to solve these problems by placing technology in inhalers is important and relevant. Then the technologies associated with the various problems need to be addressed, systematically.

This section has been restructured to include an introductory paragraph and a more systematic flow.

Interventions to improve inhaler technique

This does not fit here??

This section has been moved to the 'Patient knowledge, perceptions and behaviours' section, where we feel it is now more appropriate.

Finally, I would like to see a series of recommendations which the authors feel are hierarchical in terms of selecting devices.

A list of pragmatic recommendations has been added in a separate section. We hope this will match the objective of the review and provide practical, useful advice to aid healthcare providers in making an informed decision on inhaler devices.